# Decoding the Role of Melatonin Structure on *Plasmodium falciparum* Human Malaria Parasites Synchronization Using 2-Sulfenylindoles Derivatives

**DOI:** 10.3390/biom12050638

**Published:** 2022-04-26

**Authors:** Lenna Rosanie Cordero Mallaupoma, Bárbara Karina de Menezes Dias, Maneesh Kumar Singh, Rute Isabel Honorio, Myna Nakabashi, Camila de Menezes Kisukuri, Márcio Weber Paixão, Celia R. S. Garcia

**Affiliations:** 1Departamento de Química, Instituto de Química, Universidade de São Paulo, São Paulo 05508-000, Brazil; lenna.cordero.m@gmail.com; 2Departamento de Análises Clínicas e Toxicológicas, Faculdade de Ciências Farmacêuticas, Universidade de São Paulo, São Paulo 05508-000, Brazil; bkmdias@gmail.com (B.K.d.M.D.); manish.sings@gmail.com (M.K.S.); rute.honorio@usp.br (R.I.H.); myna@usp.br (M.N.); 3Departamento de Parasitologia, Instituto de Ciências Biomédicas, Universidade de São Paulo, São Paulo 05508-000, Brazil; 4Centro de Excelência para Pesquisa em Química Sustentável (CERSusChem), Departamento de Química, Universidade Federal de São Carlos, São Carlos 13565-905, Brazil; cmkisukuri@gmail.com (C.d.M.K.); marelloweber@gmail.com (M.W.P.)

**Keywords:** antimalarial, 2-sulfenylindoles, melatonin, *Plasmodium falciparum*, chloroquine-resistant parasites

## Abstract

Melatonin acts to synchronize the parasite’s intraerythrocytic cycle by triggering the phospholipase *C*-inositol 1,4,5-trisphosphate (PLC-IP_3_) signaling cascade. Compounds with an indole scaffold impair in vitro proliferation of blood-stage malaria parasites, indicating that this class of compounds is potentially emerging antiplasmodial drugs. Therefore, we aimed to study the role of the alkyl and aryl thiol moieties of 14 synthetic indole compounds against chloroquine-sensitive (3D7) and chloroquine-resistant (Dd2) strains of *Plasmodium falciparum*. Four compounds (**3**, **26**, **18**, **21**) inhibited the growth of *P. falciparum* (3D7) by 50% at concentrations below 20 µM. A set of 2-sulfenylindoles also showed activity against Dd2 parasites. Our data suggest that Dd2 parasites are more susceptible to compounds **20** and **28** than 3D7 parasites. These data show that 2-sulfenylindoles are promising antimalarials against chloroquine-resistant parasite strains. We also evaluated the effects of the 14 compounds on the parasitemia of the 3D7 strain and their ability to interfere with the effect of 100 nM melatonin on the parasitemia of the 3D7 strain. Our results showed that compounds **3**, **7**, **8**, **10**, **14**, **16**, **17**, and **20** slightly increased the effect of melatonin by increasing parasitemia by 8–20% compared with that of melatonin-only-treated 3D7 parasites. Moreover, we found that melatonin modulates the expression of kinase-related signaling components giving additional evidence to investigate inhibitors that can block melatonin signaling.

## 1. Introduction

Malaria is a parasitic disease of importance to public health that affects millions of people worldwide, and *Plasmodium falciparum* is responsible for most deaths from malaria that occur annually around the world. Artemisinin-based combination therapy (ACT), which consists of artemisinin, or a derivative combined with other compounds with different action mechanisms and a longer half-life, is the first-line treatment for uncomplicated malaria. However, there is a threat of parasites becoming resistant to ACT. Therefore, antimalarial treatment failure rates are monitored to avoid the high mortality rates of the 1980s, when chloroquine-resistant parasites appeared in Africa [1].

In the scenario of the global burden of the spread of chloroquine- and artemisinin-resistant *P. falciparum*, the search for new molecules to replace those used in classical malaria therapy is important. Compounds with distinct chemical architectures have shown antimalarial activity and different mechanisms of action [2,3], e.g., 4-aminoquinolines, antifolates, aryl-amino alcohols, naphthoquinones, antibiotics, and endoperoxides. In addition to traditional antimalarial drugs, other compounds have also displayed potential antiplasmodial activity; among these drugs, those containing an indole moiety are of high importance [4,5].

The host hormone melatonin is an indole derivative that synchronizes *P. falciparum* and *P. chabaudi* parasites [6]. Strikingly, the nonsynchronous murine parasites *P. berghei* and *P. yoelii* do not display melatonin-dependent synchronization in vitro [7]. Moreover, some indole derivatives, such as *N*-acetyl-serotonin and serotonin, and synthetic indole analogs also modulate parasitemia in vitro [8,9,10], and others, such as 8-oxo-tryptamine, show antiplasmodial activity against *P. falciparum* [11]. It was shown that melatonin receptor antagonist, luzindole, disrupts Ca^2+^ oscillation and cAMP increase in asexual *P. falciparum,* causing severe growth defects in intraerythrocytic maturation [12]. Other indole compounds that inhibited melatonin-induced synchronization and exhibited antimalarial activity against 3D7 parasites are the C2-arylalkanimino tryptamine derivatives [13]. These studies showed that compounds resembling the melatonin structure might disrupt the parasite life cycle. Using a rational indole synthesis strategy, several research groups have independently reported that indole derivatives are able to impair the intraerythrocytic development of *Plasmodium* [10,11,13,14,15,16,17,18,19]. Additionally, reports on 2- and 3-functionalized indoles with biological properties have led biofunctionalized indole compounds to be seen as promising bioactive molecules against malaria parasites, such as Flinderoles [20] or Borreverines [21]. In addition, among these analogs, spiroindolone KAE609 has high antimalarial activity and is currently in phase II clinical trials [22].

In this study, we aimed to investigate the antiplasmodial activity of a series of 14 structurally divergent 2-sulfenylindoles [23] against chloroquine-sensitive (CQS) 3D7 and chloroquine-resistant (CQR) Dd2 *P. falciparum* strains. In addition, we also investigated the cumulative effects of the compounds and melatonin on the parasitemia of 3D7 and Dd2 in vitro. Additional evidence of melatonin on the expression of signaling components was also investigated.

## 2. Materials and Methods

### 2.1. Plasmodium Falciparum Culture

The parasites 3D7 and Dd2 were cultured at 37 °C in RPMI-1640 medium (Gibco, Waltham, MA, USA) with 0.21% sodium bicarbonate (Sigma, Burlington, MA, USA) and 50 mg/L hypoxanthine (Sigma) supplemented with 0.5% AlbuMAX I (Gibco). The cultures were maintained under an atmosphere of 5% CO_2_, 5% O_2,_ and 90% N_2_. Fresh complete medium was provided daily, and rapid-stained (Panotico) blood smears were examined.

### 2.2. In Vitro Drug Susceptibility Assay

To evaluate the effects of synthetic compounds on the intraerythrocytic development of *P. falciparum*, we analyzed parasitemia by flow cytometry with 0.3% initial parasitemia and 1% hematocrit in the assay. Plates were incubated with the compounds at concentrations ranging from 0.10 to 100 µM for 72 h. Dual staining with SYBR Green I (SG-I) (Invitrogen, Waltham, MA, USA) and MitoTracker Deep Red (MT-Red) (Invitrogen) were used to label nucleic acids and mitochondria, respectively. Approximately 10,000 cells were counted by an Accuri C6 (Becton Dickinson, San Jose, CA, USA) and analyzed by FlowJo 5 software. Parasitemia was determined from the analysis of dot plots (side scatter versus fluorescence). DMSO-treated parasites (0.125%) were used as a negative control.

### 2.3. Cytotoxicity Assay

HEK293T (human embryonic kidney) cells were cultured in 75 cm^2^ vented tissue culture flasks (Greiner Bio-One, Frickenhausen, Germany) at 37 °C in a humidified atmosphere containing 5% CO_2_ in Dulbecco’s modified essential medium (Gibco) supplemented with 10% (*v*/*v*) fetal bovine serum (Gibco), 100 U/mL penicillin, and 100 μg/mL streptomycin (Sigma). Cytotoxicity was evaluated with the 4,5-dimethylthiazol-2-yl-2,5-diphenyltetrazolium bromide (MTT) cell proliferation assay [24,25]. Briefly, 10^4^ cells/well were seeded in flat-bottom 96-well plates in complete DMEM medium for 24 h, followed by incubation with different concentrations of each compound (0.10–100 µM) for 72 h. DMSO (0.13%) and digitonin were used as the negative and positive controls, respectively. The concentrations of DMSO chosen as control were the concentration present in the tested compounds (from 0.0001% to 0.12%, two-fold dilution). After 72 h, cells were then incubated with the MTT reagent for 3 h, and absorbance was measured at 570 nm using a FlexStation^®^ 3 Multi-Mode Microplate Reader (Molecular Devices, Sunnyvale, CA, USA). Moreover, we calculated the selectivity indexes (SI) to understand whether the compounds were more toxic to parasites than cells, or vice versa, according to the ratio of the CC_50_ value to the IC_50_ value for each compound. The data are shown in Appendix A.

### 2.4. Effects of Indole Derivatives on Parasitemia

To evaluate the role of the compounds in parasitemia, 1% 3D7 parasites in 2% hematocrit were incubated for 48 h with 500 nM of the 2-sulfenylindole derivatives. Melatonin (100 nM) was used as a positive control, and DMSO (0.0013% *v*/*v*) was used as a negative control. The concentration of DMSO chosen as control was the concentration present in the tested compounds at 500 nM. The activity of the compounds was measured by SG-I and MT-Red staining. The data obtained were normalized to those of the control (DMSO). Additionally, to test the ability of the compounds to affect melatonin activity in parasitemia, 3D7 parasites were incubated with 500 nM indole compounds along with 100 nM melatonin for 48 h.

### 2.5. Blood-Stage Development Evaluation by Microscopy

Ring-stage parasites (10–14 h) at 1% parasitemia in 2% hematocrit were treated with compound 20 and compound 28 at concentrations of 1 and 30 µM in 24-well plates for 48 h. Treatment with chloroquine (20 nM) and DMSO (0.1%) was used as positive and negative controls. Smears were rapid stained (Panotico) every 12 h after treatment (12, 24, 36, 48, and 60 h time points), and we took images of parasites to assess the blood-stage development (Zeiss, Oberkochen, Germany).

### 2.6. Real-Time PCR and Data Analysis

*P. falciparum* trophozoites (approximately 30–34 h post invasion) were treated with 100 nM and 1 µM melatonin (SIGMA-Aldrich, Burlington, MA, USA). RNA was extracted with Trizol (Life Technologies, Carlsbad, CA, USA) 1 and 3 h post treatment. cDNA synthesis was carried out using random primers and Superscript IV reverse transcriptase (Life Technologies) according to the manufacturer’s protocol. Oligonucleotides used are listed in Appendix A. SYBR Green incorporation was measured during PCR amplification performed on the 7300 Real-Time PCR System (Applied Biosystems, Foster City, CA, USA). The statistical analysis was performed by applying Student’s *t*-test on the log2 values of relative expression (delta delta Ct method) of the genes (normalized with the seryl t-RNA synthase housekeeping gene).

### 2.7. Data Analysis

The experiments were performed independently at least three times, and each experiment was analyzed in triplicate unless mentioned otherwise. Data analyses were performed with GraphPad Prism 8, and the results are expressed as the mean and standard deviation (SD) of three independent experiments. *p* < 0.05 was considered significant. One-way analysis of variance (ANOVA) followed by Dunnett’s post-hoc test was used to determine the statistical significance of the comparison of parasitemia among groups.

## 3. Results

### 3.1. Antiplasmodial Activity of 2-sulfenylindoles against CQS (3D7) and CQR (Dd2) Parasites

We determined the susceptibility of *P. falciparum* to the indole compounds and investigated whether CQR parasites responded differently to the compounds. Asynchronous parasite cultures were incubated with different concentrations of the compounds (0.1 to 100 µM) for 72 h, followed by labeling of active mitochondria in parasites with MT-Red and DNA with SG-I to identify viable parasites. We determined the IC_50_ values for each compound by analyzing viable and non-viable parasites after compound treatment, represented in dot plots (Figure 1H).

Figure 1 shows the survival curves obtained for compounds **16**, **17**, **18**, **19**, **20**, and **21** for the 3D7 and Dd2 strains. Compounds **16**, **17**, and **21** have an S-alkyl side chain attached to C2 of the indole ring, tert-butylthio, ethylthio, and dodecylthio, respectively. Compound **18** has an adamantanethio group at C2 of the indole ring, **19** has an S-alkyl chain containing a hydroxy group, and **20** has an S-alkyl chain containing a terminal carboxylic acid.

Compound **20** was the sulfenylindole studied with the most potent antiplasmodial activity against CQR. In addition, **18** and **21** suggest that a bulky group or a long alkyl side chain at C2 of the indole ring is likely important for the antiplasmodial activity of 2-sulfenylindoles (Table 1).

In Figure 2, similar survival curves were obtained using compounds **3**, **7**, **8**, **10**, **14**, **25**, **26,** and **28**. Compounds **3**, **7**, **8**, and **25** have a phenylthio group attached to C2 of the indole ring and an ester component attached to C3. Compound **10** has a pyrimidinylthio group at C2 of the indole ring, **14** has a benzylthio at C2 of the indole ring, **26** has a phenylthiol group at C2 and a benzyl group attached to C3 of the indole ring, and **28** has a phenylthiol group at C2 and an amide group attached at C3 of the indole ring.

We observed that compound **3** was the most effective against the 3D7 strain (IC_50_ = 10.58 µM). Compound **3** was more active against 3D7 parasites than its 3-acetamide counterpart (**28**, IC_50_ = 31.75 µM), and their activities against Dd2 parasites were not the same. In addition, the hydroxy substituent present in **7** had a negative effect on the antiplasmodial activity of the thioindole in vitro (Table 1).

Five compounds had IC_50_ values lower than 20 µM against Dd2 (**3**, **18**, **20**, **21**, and **28**). In addition, we found four compounds with IC_50_ values lower than 20 µM against 3D7 (**3**, **18**, **21**, and **26**). We also examined the impact of a methoxy group at C5 of the indole ring in indole analogs presenting similar substituents. Compounds **25** and **3** show that a methoxy group at C5 results in lower antiplasmodial activity, IC_50_^3D7^ = 26.47 ± 4.87 µM and IC_50_^3D7^ = 10.58 ± 5.19 µM, respectively.

In general, sulfenylindoles are more active against the CQS parasites than the CQR strain. The compounds more active against 3D7 parasites are **3**, **26**, and **18**; the two first ones are arylthioindoles, and the other one is an alkylthioindole. However, **20** and **28** are more active against CQR strain, and the first one is an alkylthioindole and the other an arylthioindole. Their resistant indexes are 0.30 and 0.39, respectively. Moreover, the selectivity index for these compounds was calculated using 100 as CC_50_ since performing a CC_50_ assay with higher concentrations could be challenging due to the compound’s solubility. Compounds **3**, **26**, and **18** exhibited low selectivity to Plasmodium parasites, while compounds **20** and **28** were modestly more selective, exhibiting selectivity index values >3.15.

### 3.2. Effect of 2-sulfenylindoles on the Parasitemia of 3D7 Parasites

Previous studies have reported that melatonin promotes parasite growth in the nanomolar range [6]. Hence, we assessed the effect of compounds on parasitemia by incubating asynchronous 3D7 parasites with the set of 2-sulfenylindoles at 500 nM for 48 h. After incubation, the final parasitemia was determined using flow cytometry. The results are shown in Figure 3. Compounds **7** and **26** exhibited slight inhibitory effects on parasite growth, as parasitemia decreased by 4.05% and 7.94%, respectively. Moreover, compounds **16**, **20**, **21**, **25**, and **28** led to slight increases in parasitemia of 11.2%, 9.1%, 3.8%, 6.6%, and 8.7%, respectively, compared to the control group treated with DMSO (100.6%).

Our results showed that 2-sulfenylindoles could inhibit intraerythrocytic development of parasites in vitro after 72 h incubation at the micromolar concentrations. At a lower range and after 48 h incubation, five of the compounds (**16**, **20**, **21**, **25**, and **28**) led to an increase in parasitemia, in a similar way as that observed for the host hormone melatonin (a rise of 6.9% from 100.6% to 107.5%) and its derivatives [8].

### 3.3. Effect of 2-sulfenylindoles in Combination with Melatonin on the Parasitemia

Following the assumption that compounds resembling melatonin structure might interact through the melatonin signaling pathways, we evaluated the effect of the compounds on melatonin action on parasitemia. Our next step was to study the combination of 2-sulfenylindoles with melatonin in asynchronous 3D7 by incubating parasites simultaneously with 100 nM melatonin and 500 nM of each compound.

The combination of compound **3** (500 nM) with melatonin (100 nM) had a synergistic effect, enhancing parasitemia by 16.27% compared with that of parasites exposed only to 100 nM melatonin (113.5%). Similar responses were observed for compounds **8**, **14**, **17**, **10**, **7**, **16,** and **20** (Figure 4). After simultaneous exposure to melatonin and compound **8**, the parasitemia of 3D7 parasites increased by 20% compared to that of melatonin-only-treated parasites. Similarly, a synergistic effect on parasitemia was observed after incubation with melatonin along with some compounds, leading to an increase of 16.63% (**10**), 14.88% (**14**), 13.88% (**16**), 10.05% (**17**), 9.42% (**7**), and 8.81% (**20**) (Appendix A).

As mentioned above, at micromolar concentrations, we observed an inhibitory effect for each compound. However, compound **7** (500 nM) alone slightly decreased parasitemia by 4.084%, whereas in combination with melatonin, it increased parasitemia by 22.2% compared to that of the control group (100.7%).

### 3.4. Cytotoxic Activity of 2-sulfenylindoles on Mammalian Cells

We further assessed the toxicity of 2-sulfenylindole compounds against HEK293 cells using the MTT assay with different concentrations of the compounds for 72 h. The data are shown in Appendix A. Table 1 presents all the toxicity data and summarizes the IC_50_ values of the compounds evaluated in this study for Dd2 and 3D7 strains. All the compounds exhibited IC_50_ values in the micromolar range.

Compound **25** was the least non-toxic with a CC_50_ of 13.14 ± 3.74 µM in HEK293 cells, whereas compound **21** was the least toxic with a CC_50_ of >100 µM (Appendix A). Some of the 2-sulfenylindoles displayed activity against HEK293 cells, and selectivity indexes to *Plasmodium* parasites over mammalian cells of between 0.28 and 2.58 times were identified (Table 1). The CC_50_ values for five compounds were not determined, and we speculated that these compounds exhibit more selective toxicity to the *Plasmodium* parasites than to HEK293 cells because the highest test concentration was not able to kill 100 of the mammalian cells and is impossible to evaluate higher concentrations due to compounds solubility. Therefore, we considered that compounds 8, 10, 20, 21, and 28 exhibit more selective toxicity to the *Plasmodium* parasites than to HEK293 cells.

### 3.5. Effect on Blood-Stage Growth Progression of the Most Active Compounds

Analyses with microscopy reveal the blood-stage development inhibition in 3D7 parasites treated with compounds **20** and **28** at IC_50_ (Figure 5C) and compound **3**, **18**, and **26** at IC_50_ (Appendix A). During monitoring, parasites continued to develop throughout the cycle in the presence of both compounds. However, compounds **20** and **28** significantly affect parasitemia (Figure 5A,B, respectively). To examine its stage specificity, we monitored development every 12 h. Compound **28** had no effect on parasite growth at the schizont stage (36–48 h time points) but stalled ring to trophozoite progression. On the other hand, compound **20** affects RBC reinvasion, evidenced by the reduction in ring formation at the 48 h time point.

Treatment with compounds **3**, **18**, and **26** allowed parasites’ development continues. However, in the second cycle of proliferation, these compounds reduced the progress from rings to trophozoites (Appendix A).

### 3.6. Indole Compound Melatonin Alters the Expression of Plasmodium Transcript

Previous studies have shown that melatonin alters the expression of the ubiquitin/proteasome system [26], mitochondrial FIS1, DYN2 [27], and nuclear PfMORC [28]. These findings lead us to investigate the effect of melatonin on kinases, phosphodiesterase, and cyclase for two reasons; first, kinases and their components are required for rupturing the parasites to facilitate egress in both asexual and sexual stages. Second, in the asexual blood stage, melatonin has been shown to mobilize calcium from internal stores [29,30], and prior to egress, calcium is mobilized by PKG-mediated breakdown of PIP2 [31]. We treated the late-stage trophozoite with two different melatonin concentrations (100 nM and 1 µM) for two time points (1 and 3 h). In our result (Figure 6), we found that 100 nM melatonin for 1 h has a very minor effect on change in gene expression for CDPK1 and GCα; however, after 3 h, increased expression can be seen in additional CDPK6, CDPK7, PKG, PDEα, and GCβ. Interestingly, 1 µM melatonin for 3 h treatment has the most prominent effect on change in gene expression for cGMP-dependent signaling component suggesting melatonin role in the asexual cycle. It is possible that melatonin may act upstream of PKG and modulates cytosolic calcium, but this hypothesis requires more experimental validation.

## 4. Discussion

The parasite maturation during the intraerythrocytic cycle is extremely coordinated in which schizonts rupture and invade new erythrocytes in a highly synchronized manner at intervals usually multiples of 24 h [32,33,34]. Our lab has shown that the developmental synchrony of the parasite was lost in pinealectomized mice [6]. Calcium is well known to be central in controlling the molecular processes of the cell cycle in parasites [35,36]. The downstream molecular mechanism of melatonin involves the PLC-IP_3_ signaling pathways [30,37,38] through increases of Ca^2+^ and cAMP, leading to the activation of PKA [8,39].

Several studies have shown that melatonin modulates the cell cycle of the human malaria parasite *P. falciparum*, determining the synchrony of the invasion of red blood cells by intraerythrocytic *Plasmodium* parasites and their asexual reproduction [40,41]. Therefore, the potential for using melatonin derivatives containing an indole moiety to treat malaria has been reported previously [10,13,19].

Antiplasmodial compounds containing a benzenesulfonamide moiety [19] or a carboxamide group [10] at the C3 position of the indole ring have also been reported. Additionally, some marine-indole alkaloids with substitutions at the C3-position of the indole ring are active against *P. falciparum*, with IC_50_ values of 4.44 to 200 µM. In addition, evidence suggests that bromide and hydroxyl substitutions at the C7 and C4 positions of the indole ring, respectively, increase activity against parasites [16,17].

In our study, sulfenyl indoles showed inhibitory activity ranging from 8 to 59 µM for both 3D7 and Dd2 parasites. Compound **21** is a 6-chloro analog of meridoquin (IC_50_ = 200 µM) [16] and showed modest antiplasmodial activity compared to that of 2-sulfenyl indoles in our study, suggesting that the presence of the S-alkyl chain at the C2 position of the indole ring resulted in greater improvement of antiplasmodial activity than the presence of a pyrimidine group at C3 of the indole ring.

We observed that compounds **20** and **28**, which had significantly lower IC_50_ values against the Dd2 strain, had a low cytotoxic effect against HEK293 cells. In addition, these compounds alone at a nanomolar concentration (500 nM) were able to significantly increase the parasitemia of 3D7 asynchronous cultures, indicating that although they can increase parasitemia at low concentrations, they can selectively affect parasite growth at higher concentrations. Previous studies have reported that a 4-aminoquinoline-piperidine is less effective against a wild-type NF54 strain than against a CQR strain K1 [42]. Similarly, some chloroquine analogs have been shown to have higher activity against W2 than 3D7 parasites [43]. Furthermore, β-benzoylstyrene derivatives of acridine with better potency against Dd2 than the 3D7 have been reported [44]. Compounds with better activity against CQR strains are a promising avenue of research, as reported by Ngemenya et al. [45].

It has been reported that an indole compound with a methoxy group at C5, melatotosil, can increase parasitemia and interfere with the action of melatonin [19]. Our results also suggested that the presence of a methoxy group at C5 in compound **25** led to a slight increase in parasitemia of 6.66%, while no increase was observed after exposure to its counterpart (compound 3). However, treatment with compound **25** in combination with melatonin did not potentiate or block the effect of melatonin on parasitemia. On the other hand, a slight increase in parasitemia was observed after exposure to compound **3** in combination with melatonin compared with that after exposure to melatonin alone (113.5%). Compounds **25** and **3**, unlike melatotosil, showed antimalarial activity (Table 1).

Since compounds **16** and **21** were able to slightly increase parasitemia at 500 nM and since both have an S-alkyl group at C2 of the indole ring, a small alkyl substituent might favor an increase in parasitemia since compound **16** contains a tert-butylthio group and led to an increase in parasitemia of 11.2%. Compound **21**, which has a long alkyl side chain, led to an increase in parasitemia of 3.8%. However, a compound with a smaller alkyl group (compound **17**) did not affect parasitemia.

Moreover, real-time data with melatonin implicates the alteration in kinases and their components related to egress and/or invasion. It is more evident that the development of more indole-related compounds may exhibit an antimalarial effect by affecting genes that modulate both egress and invasion.

## 5. Conclusions

Two compounds have shown slightly more activity on the resistant Dd2 strain with lower IC_50_ values; these results suggest the potential of sulfenylindoles as antiplasmodial. Compounds **3**, **18**, **26**, and **28** affect ring to trophozoite progression, whereas compound **20** affects RBC reinvasion. In addition, aiming to obtain a compound that blocks parasite development, we evaluated the action of 14 compounds on parasitemia at nanomolar concentrations. Compound **3**, which has a phenylthio group at C2 of the indole ring, presented the lower IC_50_ value in 3D7 (IC_50_ = 10.58 µM). However, the response was not the same against Dd2 parasites. Compound 20, an S-alkyl chain at C2, was the most active against Dd2 parasites (IC_50_ = 8.72 µM). Moreover, five indole compounds (**16**, **20**, **21**, **25**, and **28**) increase parasitemia at low concentrations, whereas at high ones, they reduce parasitemia. These findings suggest sulfenylindoles as potential antimalarials against parasites resistant and susceptible to chloroquine despite some of the compounds tested can increase parasitemia, similar to the host hormone melatonin.

## Figures and Tables

**Figure 1 biomolecules-12-00638-f001:**
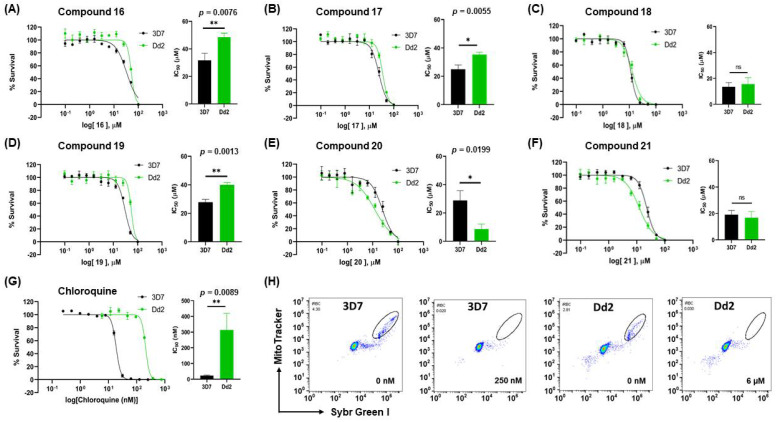
Determination of the antiplasmodial activity of 2-sulfenyl indoles in the 3D7 and Dd2 strains. Asynchronous parasites were exposed to compounds **16**, **17**, **18**, **19**, **20**, and **21** for 72 h in concentrations ranging from 0.1 to 100 µM. Parasitemia was calculated using a flow cytometer by double staining parasites with SYBR and MitoTracker. Dot plot analysis of each drug concentration tested allowed us to determine parasitemia. Growth survival curves of blood-stage *P. falciparum* of compounds **16** (**A**), **17** (**B**), **18** (**C**), **19** (**D**), **20** (**E**), and **21** (**F**) for 3D7 (black) or Dd2 (green). Chloroquine (**G**) was administered at concentrations from 0 to 250 nM to 3D7 parasites and from 0 to 6 µM to Dd2 parasites. Data represent three independent experiments performed in triplicate. (**H**) Dot plots representing SG-I/MT-Red staining of control and chloroquine-treated asynchronous 3D7 (left) and Dd2 (right) parasites after 72 h of treatment. The bar graphs represent the SD. Statistical significance was calculated using Student’s *t*-test: ns = not significant, * *p* ≤ 0.05, ** *p* ≤ 0.01.

**Figure 2 biomolecules-12-00638-f002:**
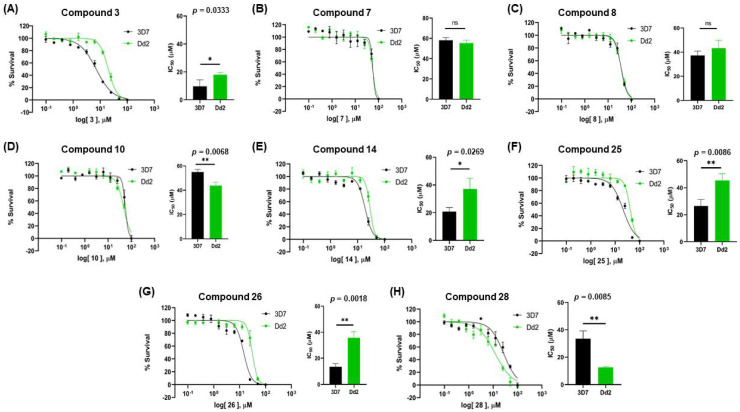
Determination of the antiplasmodial effects of 2-sulfenyl indoles on the 3D7 and Dd2 strains. Asynchronous parasites exposed to compounds **3**, **7**, **8**, **10**, **14**, **25**, **26,** and **28** for 72 h in the range of 0.1–100 µM. Parasitemia was obtained using a flow cytometer by double staining parasites with SYBR and MitoTracker. Dot plot analysis of each drug concentration tested allowed us to determine parasitemia. Growth survival curves of blood-stage *P. falciparum* of compounds. Growth survival curves of blood-stage *P. falciparum* of compounds **3** (**A**), **7** (**B**), **8** (**C**), **10** (**D**), **14** (**E**), **25** (**F**), **26** (**G**), and **28** (**H**) for 3D7 (black) or Dd2 (green). Data represent three independent experiments performed in triplicate. The bar graphs represent the SD. Statistical significance was calculated using Student’s *t*-test: ns = not significant, * *p* ≤ 0.05, ** *p* ≤ 0.01.

**Figure 3 biomolecules-12-00638-f003:**
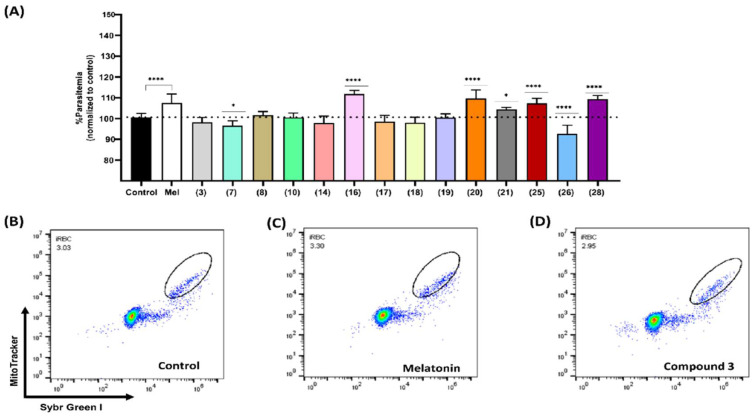
Effects of 2-sulfenyl indoles on the parasitemia of *P. falciparum* after 48 h of incubation with 500 nM compounds **3**, **7**, **8**, **10**, **14**, **16**, **17**, **18**, **19**, **20**, **21**, **25**, **26**, and **28** (**A**). Dot plots showing SG-I/MT-Red-labeled parasites detected by flow cytometry after 48 h of treatment with DMSO (**B**), 100 nM melatonin (**C**), and 500 nM compound 3 (**D**). Data are presented as the percentage of parasitemia normalized to that of the control group treated with solvent (DMSO). Experiments were performed 3 independent times in triplicate. The error bars represent the SD. * indicates a significant difference from the control group. Statistical significance was calculated using one-way ANOVA followed by Dunnett’s test: * *p* ≤ 0.05, **** *p* ≤ 0.0001.

**Figure 4 biomolecules-12-00638-f004:**
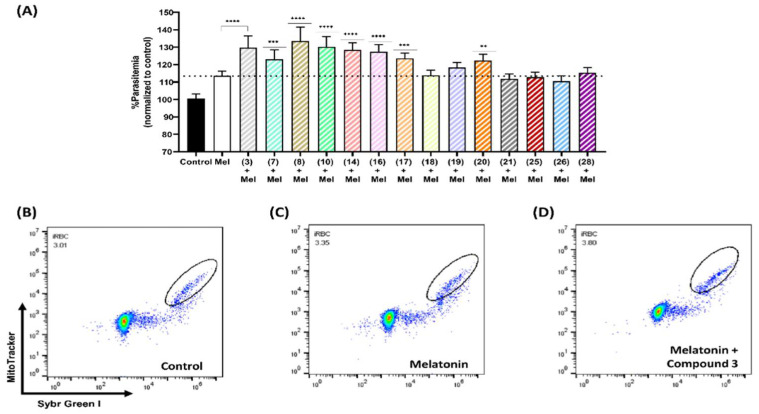
Effects of 2-sulfenyl indoles on the parasitemia of *P. falciparum* after 48 h of incubation with 500 nM compounds **3**, **7**, **8**, **10**, **14**, **16**, **17**, **18**, **19**, **20**, **21**, **25**, **26**, and **28** in combination with 100 nM melatonin (Mel) (**A**). Dot plots showing SG-I/MT-Red-labeled parasites detected by flow cytometry after 48 h of treatment with DMSO (**B**), 100 nM melatonin (**C**), and a combination of 100 nM melatonin and 500 nM compound 3 (**D**). Data are presented as the percentage of parasitemia normalized to that of the control group treated with solvent (DMSO). Experiments were performed 3 independent times in triplicate. The error bars represent the SD. * indicates a significant difference compared to the melatonin treatment (Mel). Statistical significance was calculated using one-way ANOVA followed by Dunnett’s test: ** *p* ≤ 0.01, *** *p* ≤ 0.001, **** *p* ≤ 0.0001.

**Figure 5 biomolecules-12-00638-f005:**
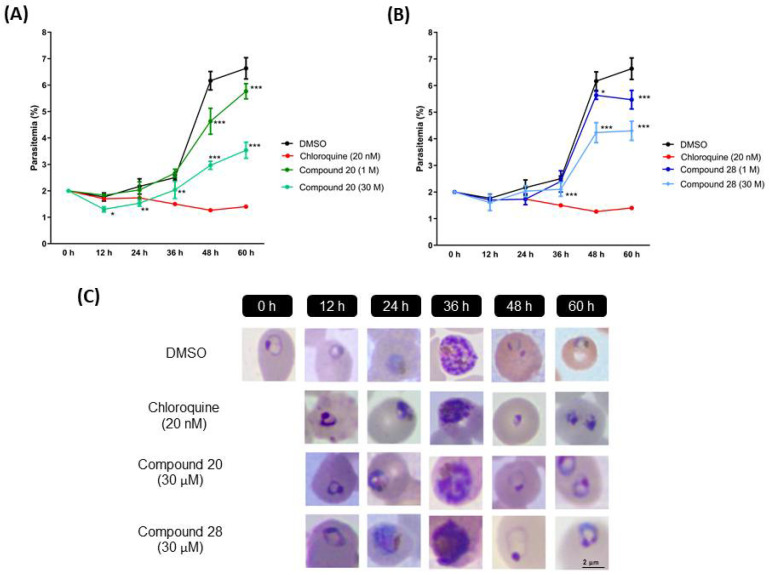
Parasitemia reduction with compound **20** (**A**) and compound **28** (**B**) to determine the stage activity. Data are means ± SD. Ring-stage synchronized parasites with sorbitol 5% were treated with chloroquine (20 nM), compound **20** (30 µM), and compound **28** (30 µM). The chosen treatment concentration was the IC_50_ values obtained for these compounds against the 3D7 parasites. (**C**) After drug administration, smears of *P. falciparum* 3D7 strain were taken at 0, 12, 24, 36, 48, and 60 h after drug administration, observing the growth development of treated parasites. * *p* ≤ 0.05, ** *p* ≤ 0.01, *** *p* ≤ 0.001.

**Figure 6 biomolecules-12-00638-f006:**
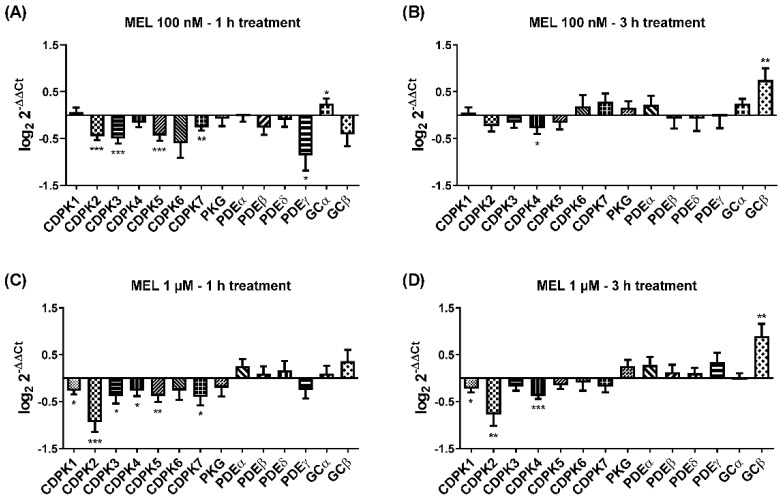
Differential transcription of CDPKs, PKG, and its accessory genes from parasites treated with melatonin for 1 and 3 h. Sorbitol synchronized parasites were treated with melatonin, and RNA samples were extracted, purified, and transcript level was investigated by RT-qPCR. Expression of genes has been normalized by housekeeping seryl-tRNA Synthetase expression for (**A**) 100 nM melatonin for 1 h; (**B**) 100 nM melatonin for 3 h; (**C**) 1 μM melatonin for 1 h; and (**D**) 1 μM melatonin for 3 h. Each figure represents three independent experiments in triplicate. * *p* ≤ 0.05, ** *p* ≤ 0.01, *** *p* ≤ 0.001.

**Table 1 biomolecules-12-00638-t001:** In vitro activity of the compounds against asexual blood stages of *Plasmodium falciparum* and cytotoxicity in the HEK293 cell line.

Compound	IC_50_ ± SD *Pf3D7*	IC_50_ ± SD *PfDd2*	CC_50_ ± SD HEK293	SI 3D7	SI Dd2
(**3**) 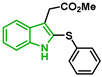	10.58 ± 5.19 µM	17.97 ± 1.72 µM	15.24 ± 1.36 µM	1.44	0.84
(**7**) 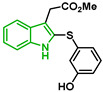	58.18 ± 2.45 µM	55.42 ± 2.96 µM	31.59 ± 4.03 µM	0.54	0.57
(**8**) 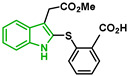	37.29 ± 3.58 µM	43.29 ± 6.51 µM	>100 µM	>2.68	> 2.31
(**10**) 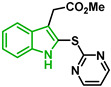	54.86 ± 2.25 µM	43.62 ± 3.060 µM	>100 µM	>1.82	> 2.29
(**14**) 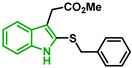	20.86 ± 2.97 µM	37.08 ± 7.664 µM	23.69 ± 8.540 µM	1.14	0.63
(**16**) 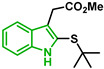	31.69 ± 5.09 µM	48.50 ± 2.88 µM	26.52 ± 8.70 µM	0.84	0.55
(**17**) 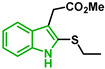	24.99 ± 2.90 µM	35.37 ± 1.56 µM	26.49 ± 4.67 µM	1.06	0.75
(**18**) 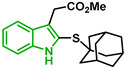	13.52 ± 3.24 µM	15.72 ± 4.87 µM	16.82 ± 2.63 µM	1.24	1.07
(**19**) 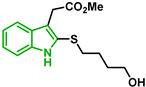	27.78 ± 2.03 µM	39.92 ± 1.61 µM	28.14 ± 5.46 µM	1.01	0.70
(**20**) 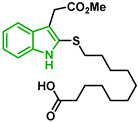	28.78 ± 4.86 µM	8.72 ± 3.36 µM	>100 µM	>3.47	>11.47
(**21**) 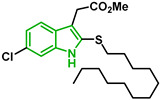	19.10 ±1.78 µM	16.86 ± 4.66 µM	>100 µM	>5.24	>5.93
(**25**) 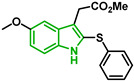	26.47 ± 4.87 µM	45.53 ± 4.84 µM	13.14 ± 3.74 µM	0.50	0.28
(**26**) 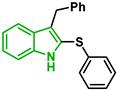	13.48 ± 2.37 µM	35.80 ± 4.64 µM	34.72 ± 8.44 µM	2.58	0.97
(**28**) 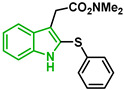	31.75 ± 5.02 µM	12.39 ± 0.59 µM	>100 µM	>3.15	>8.07
**Chloroquine**	22.33 ± 4.17 nM	199.05 ± 26.23 nM	ND	ND	ND

ND: not determined, SI: selectivity index (CC_50_ value of HEK293 cell/IC_50_ value of Pf3D7 or PfDd2).

## Data Availability

Not applicable.

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
