# Peer review of "Decoding the Role of Melatonin Structure on Plasmodium falciparum Human Malaria Parasites Synchronization Using 2-Sulfenylindoles Derivatives"

_biomolecules, 2022, doi:10.3390/biom12050638_

Round 1
Reviewer 1 Report
Cordero-Mallaupoma and colleagues report the antimalarial activity and selectivity index of fourteen synthetic compounds with an indole scaffold as melatonin. Acknowledging previous melatonin effect on parasitemia and intraerythrocytic stage synchrony, they have also studied theses effect on the synthetic compounds. As such it is interesting and has novelty. The usefulness of the finding regarding the parasite transcripts when exposed to melatonin though, are not so clear and confuses the main message of the work towards the potential of new indole compounds as novel antimalarial drugs. In this regard, considering how results are presented and respective discussion section, does not reflect the chosen title. There is a lack of structure comparison between the synthetic compounds and melatonin that preclude accomplish the proposed manuscript title. In this regard, why was transcripts only studied under melatonin exposure, discarding the synthetic compounds?
Other comments:
- Why decided to perform drug susceptibility assay in asynchronous cultures? Considering the growth assay performed (figure 5), the most promising compounds should be tested in synchronous cultures at different stages. This will allow to define the stage where the compounds are most active.
- What’s the rational for separating figure 1 from figure 2?
- Table 1 suggest to add melatonin structure for comparison given authors aim of “decoding the role of melatonin structure”. In this sense, melatonin CC50 and IC50 should also be provided.
- Authors should discuss literature SI threshold and position their best compounds. SI was not calculated for compounds with CC50>100uM. I suggest to calculate using 100 as CC50 with resulted SI presented as not equal (=) but above (>) the obtained value. For best compounds with CC50>100uM, I would also suggest to perform new CC50 assay with at least one more log of drug concentration. Hopefully, this will allow to increase the SI.
- Line 321: “compound 20 affects RBC reinvasion, evidenced by the reduction of ring formation at the 48-hour time point.”. from figure 5 we can observe that ring formation from 0 to 12 hours the line has the same tilt as of 48-60 hours for the compound 20 being difficult to conclude a reduction of ring formation. This conclusions also emphasize the need to perform drug susceptibility assay at specific intraerythrocytic stages as commented above.
- Figure 5A and 5B shows no growth when parasites are exposed to chloroquine, nevertheless Figure 5C shows images of parasite progression on all stages when exposed to chloroquine. Figure 5C legend is missing.
Other minor comments:
- Line 80: describe the source of the reagents. AlbuMAX used was I or II?
- Line 113: describe CC50.
- Line 418: rephrase “realtime”.
- All figures are too small, some difficult to read.
Author Response
São Paulo, April 11th, 2022.
Prof. Dr. Vladimir N. Uversky
Editor-in-Chief,
Biomolecules
Dear Prof. Uversky
Please find the on line submission of the revised version of manuscript: Decoding the role of melatonin structure on Plasmodium falciparum human malaria parasites synchronization using 2‐sulfenylindoles derivatives by Lenna R. Cordero-Mallaupoma, Barbara Karina Dias, Maneesh Kumar Singh, Rute Isabel Honório, Myna Nakabashi, Camila Kisukuri, Marcio Weber Paixão and myself that we are submiting for publication at Biomolecules.
We are grateful to the reviewers’s criticism as the revised version of the manuscript is now much improved. We have performed additional experiments and modified the manuscript. Bellow is our detailed response to their queries.
We hope that you will find the revised manuscript suitable for publication at J Biomolecules.
Sincerely,
Celia R. S. Garcia, PhD
Professor
Department of Clinical and Toxicological Analyses
School of Pharmaceutical Sciences, University of São Paulo
Brazil
Reviewer 1
|
Comments and Suggestions for Authors
Cordero-Mallaupoma and colleagues report the antimalarial activity and selectivity index of fourteen synthetic compounds with an indole scaffold as melatonin. Acknowledging previous melatonin effect on parasitemia and intraerythrocytic stage synchrony, they have also studied these effects on the synthetic compounds. As such it is interesting and has novelty. The usefulness of the finding regarding the parasite transcripts when exposed to melatonin though, are not so clear and confuses the main message of the work towards the potential of new indole compounds as novel antimalarial drugs. In this regard, considering how results are presented and respective discussion sections, does not reflect the chosen title. There is a lack of structure comparison between the synthetic compounds and melatonin that preclude accomplishing the proposed manuscript title. In this regard, why were transcripts only studied under melatonin exposure, discarding the synthetic compounds?
We thank the reviewer´s suggestion, but we respectfully disagree. The ms deals with both aspects as a competition assay with melatonin is critical to evaluate the hormone´s action in the presence of a sulphenyl indole series. Moreover, we want to decode the signaling pathways of melatonin action on parasite synchronization and not follow if sulphenyl indole series compounds lead to the alteration of gene expression in Plasmodium falciparum.
Decoding the downstream signaling pathways involved in melatonin action on synchronization of malaria parasites is an important event. We have previously reported the relevance of the following kinases in this process: PKA (Beraldo et al., J Cell Biology, 2005), PfPk7 (Koyama et al., J Pineal Research, 2012) PfeIK1 (Dias et al., J. Pineal Research, 2020). As mentioned in the ms the idea here was investigating the effect of melatonin on kinases, phosphodiesterase, and cyclase. This is an area of intense research in malaria parasites and is well known that kinases and their components are required for rupturing the parasites to facilitate parasites to egress for the red blood cells. Moreover, melatonin has been shown to mobilize calcium from internal stores at blood stages. We treated the late-stage trophozoites with two different melatonin concentrations (100 nM and 1 µM) for two-time points (1 h and 3 h). We found that 100 nM melatonin for 1 h has a minor effect on change in gene expression for CDPK1 and GCα; after 3 h, increased expression can be seen in additional kinases: CDPK6, CDPK7, PKG as well as PDEα and GCβ. This is critical data, and we are preparing to send the raw data (as supplement material) and extend this discussion in the ms.
Other comments:
- Why decided to perform drug susceptibility assay in asynchronous cultures? Considering the growth assay performed (figure 5), the most promising compounds should be tested in synchronous cultures at different stages. This will allow us to define the stage where the compounds are most active.
Thanks for the reviewer suggestion. To determine the IC50 value of our sulfenylindoles, we used a 72h assay. We chose to incubate asynchronous parasites with our compounds for 72h because we can observe the effects of our compounds over a complete asexual cycle.
Later, we tested the compounds with the lowest IC50 values against CQR parasites (compounds 20 and 28) to determine the stage where the compound shows more activity (Figure 5). Following the reviewer´s suggestion, we performed the same experiment for compounds with the lowest IC50 against CQS parasites (compounds 3, 18 and 26) (Figure S2).
- What is the rationale for separating figure 1 from figure 2?
In figure 1, compounds are alkylthiols and they are arylthiols in figure 2.
- Table 1 suggests adding melatonin structure for comparison given author’s aim of “decoding the role of melatonin structure”. In this sense, melatonin CC50and IC50 should also be provided.
2-sulfenylindoles are not a typical pharmacophore, but they have an indole ring that resembles melatonin and a thioether linkage present in some bioactive molecules (La Regina et al., J. Med. Chem, 2015) (Grime et al., Br. J. Clin. Pharmacol, 2016). On the other hand, Hotta et al. (2000) described melatonin as a compound able to accelerate the intraerythrocytic developmental cycle in 3D7 parasites (Hotta et al., Nat Cell Bio, 2000).
- Authors should discuss literature SI threshold and position their best compounds. SI was not calculated for compounds with CC50>100uM. I suggest to calculate using 100 as CC50with resulted SI presented as not equal (=) but above (>) the obtained value. For best compounds with CC50>100uM, I would also suggest to perform new CC50 assay with at least one more log of drug concentration. Hopefully, this will allow to increase the SI.
CC₅ₒ values for five compounds were not determined, and we speculated that these compounds exhibit more selective toxicity to the Plasmodium parasites than to HEK293 cells because the highest concentration tested was not able to kill 100 percent of the mammalian cells, and evaluate higher concentrations could be challenging (compounds solubility). We have improved the description of our study thanks to your suggestion; however, it was not possible to perform other cytotoxicity assay.
- Line 321: “compound 20 affects RBC reinvasion, evidenced by the reduction of ring formation at the 48-hour time point.”.From figure 5 we can observe that ring formation from 0 to 12 hours the line has the same tilt as of 48-60 hours for the compound 20 being difficult to conclude a reduction of ring formation. These conclusions also emphasize the need to perform drug susceptibility assay at specific intraerythrocytic stages as commented above.
The first interval evaluated (from time point 0 to 12) allow us to point the effect of compound 20 on the progression of rings to trophozoites. Our results show a slight significance for the parasites treated with the compound 20 at 30µM. However, in the second cycle of the parasite (from time point 48 to 60), it is observed that despite the reduction in parasitemia, the line has a similar slope both for treated parasites and for control. On the other hand, the time interval that includes the invasion of new erythrocytes (from time point 36 to 48) show lines with different slopes.
- Figure 5A and 5B shows no growth when parasites are exposed to chloroquine, nevertheless Figure 5C shows images of parasite progression on all stages when exposed to chloroquine. Figure 5C legend is missing.
At the end of the incubation time, parasitemia was approximately 1% in the wells with chloroquine-treated parasites.
Figure 5 legend mentioned that Figure 5C are the smears of 3D7 parasites.
Other minor comments:
- Line 80: describe the source of the reagents. AlbuMAX used was I or II?
We used AlbuMAX I.
- Line 113: describe CC50.
Following the reviewer’s suggestions, we have improved the description of our study.
CC50 defines the 50% cytotoxic concentration of indole compounds in HEK293 cells.
- Line 418: rephrase “realtime”.
We thank the reviewer’s suggestion, real-time.
- All figures are too small, some difficult to read.
We thank the reviewer for the suggestion, and we will modify them.

Reviewer 2 Report
1. The present research is of interest due to the high morbidity and mortality of malaria.
2. The chemical development and pharmacological testing of new molecules with antimalarial potential, especially on chloroquine-resistant or resistant forms of artemisinin and derivatives are priority directions of pharmaceutical and medical research.
3. The research hypothesis, the methodology, discussions and conclusions of this study are correct
4. I would suggest, in addition to those mentioned in the previous report, to increase the size of the figures. In this form, it is difficult to verify the accuracy of the results of the experiment.5. In my opinion, the article should have numbered the compounds with 1-14. It was not necessary to keep the numbering from the article in which the compounds were reported. I also recommend to specify how the test compounds were selected from the larger group of compounds reported (in reference 41).
Even in these conditions, I recommend the publication.
Author Response
São Paulo, April 11th, 2022.
Prof. Dr. Vladimir N. Uversky
Editor-in-Chief,
Biomolecules
Dear Prof. Uversky
Please find the on line submission of the revised version of manuscript: Decoding the role of melatonin structure on Plasmodium falciparum human malaria parasites synchronization using 2‐sulfenylindoles derivatives by Lenna R. Cordero-Mallaupoma, Barbara Karina Dias, Maneesh Kumar Singh, Rute Isabel Honório, Myna Nakabashi, Camila Kisukuri, Marcio Weber Paixão and myself that we are submiting for publication at Biomolecules.
We are grateful to the reviewers’s criticism as the revised version of the manuscript is now much improved. We have performed additional experiments and modified the manuscript. Bellow is our detailed response to their queries.
We hope that you will find the revised manuscript suitable for publication at J Biomolecules.
Sincerely,
Celia R. S. Garcia, PhD
Professor
Department of Clinical and Toxicological Analyses
School of Pharmaceutical Sciences, University of São Paulo
Brazil
Reviewer 2
Comments and Suggestions for Authors
- The present research is of interest due to the high morbidity and mortality of malaria.
- The chemical development and pharmacological testing of new molecules with antimalarial potential, especially on chloroquine-resistant or resistant forms of artemisinin and derivatives are priority directions of pharmaceutical and medical research.
- The research hypothesis, the methodology, discussions and conclusions of this study are correct
- I would suggest, in addition to those mentioned in the previous report, to increase the size of the figures. In this form, it is difficult to verify the accuracy of the results of the experiment.
We thank the reviewer’s suggestion; we increased the size of the figures.
- In my opinion, the article should have numbered the compounds with 1-14. It was not necessary to keep the numbering from the article in which the compounds were reported. I also recommend specifying how the test compounds were selected from the larger group of compounds reported (in reference 41). Even in these conditions, I recommend the publication.
We appreciate the reviewer's suggestion; we decided to keep the previously reported numbering to facilitate the search for the synthesis and characterization of the compound in case other research groups are interested in studying these compounds.
We determined the solubility of the sulfenylindoles described by Santos (Santos et al., Org. Lett., 2020) in parasite culture media (RPMI). Only 14 soluble indole derivatives were selected, the other compounds presented poor solubility in RPMI media.

Round 2
Reviewer 1 Report
Figure 5C legend should describe the staining used for the smear presented and the scale-bar of the images. "3D7 parasites" should be P. falciparum 3D7 strain.
No further comments.
Author Response
São Paulo, April 12th, 2022.
Prof. Dr. Vladimir N. Uversky
Editor-in-Chief,
Biomolecules
Dear Prof. Uversky
Please find the on line submission of the revised version of manuscript: Decoding the role of melatonin structure on Plasmodium falciparum human malaria parasites synchronization using 2‐sulfenylindoles derivatives by Lenna R. Cordero-Mallaupoma, Barbara Karina Dias, Maneesh Kumar Singh, Rute Isabel Honório, Myna Nakabashi, Camila Kisukuri, Marcio Weber Paixão and myself that we are submiting for publication at Biomolecules.
We are grateful to the reviewers’s criticism as the revised version of the manuscript is now much improved. We have performed additional experiments and modified the manuscript. Bellow is our detailed response to their queries.
We hope that you will find the revised manuscript suitable for publication at J Biomolecules.
Sincerely,
Celia R. S. Garcia, PhD
Professor
Department of Clinical and Toxicological Analyses
School of Pharmaceutical Sciences, University of São Paulo
Brazil
Reviewer
Comments and Suggestions for Authors
Figure 5C legend should describe the staining used for the smear presented and the scale-bar of the images. "3D7 parasites" should be P. falciparum 3D7 strain.
Thanks for your suggestion. The suggestion has been incorporated in the ms.